# Psychological well-being in Europe after the outbreak of war in Ukraine

The Russian invasion of Ukraine on February 24, 2022, has had devastating effects on the Ukrainian population and the global economy, environment, and political order. However, little is known about the psychological states surrounding the outbreak of war, particularly the mental well-being of individuals outside Ukraine. Here, we present a longitudinal experience-sampling study of a convenience sample from 17 European countries (total participants = 1,341, total assessments = 44,894, countries with >100 participants = 5) that allows us to track well-being levels across countries during the weeks surrounding the outbreak of war. Our data show a significant decline in well-being on the day of the Russian invasion. Recovery over the following weeks was associated with an individual's personality but was not statistically significantly associated with their age, gender, subjective social status, and political orientation. In general, well-being was lower on days when the war was more salient on social media. Our results demonstrate the need to consider the psychological implications of the Russo-Ukrainian war next to its humanitarian, economic, and ecological consequences.

On February 24, 2022, Russia invaded Ukraine, thereby violating the country's territorial sovereignty and escalating the Russo-Ukrainian war that began in 2014[1,2]. Besides devastating effects on the Ukrainian population, the invasion has had severe global consequences; for example, the war has resulted in Europe's fastest-growing refugee crisis since World War II, global food shortages, and negative effects on the world economy[3–7]. The UN Global Crisis Response Group estimates that 1.6 billion people in 94 countries are exposed to at least one dimension of the crisis[8]. While effects such as the displacement of millions of civilians or disrupted supply chains are immediately visible, the psychological implications of the outbreak of war may be more difficult to trace, with potentially even more people worldwide experiencing psychological distress and impaired mental health during the war. Here, we used international, longitudinal data from January to April 2022, to investigate (a) the development of individuals' well-being during the weeks surrounding the outbreak of war, (b) potential explanations for individual differences in reactions to the event, and (c) the correspondence between daily well-being and the daily salience of the war on social media during this period.

When investigating the psychological effect of major life events such as an outbreak of war, researchers often face the problem that people volunteer for studies only after experiencing the event. Besides potentially introducing retrospective biases, such cross-sectional data do not allow for investigations of pre-post comparisons or change over time, such as anticipation or recovery effects[9,10]. Panel studies that track participants across time circumvent these limitations, but they often have a low time resolution with only one or a few assessments per year, and they are typically undertaken in a single country at a time. These limitations make it difficult to study events that presumably have their most direct psychological effects right when they happen and that affect individuals across nations (e.g., the outbreak of a war). Here, we present a study that allowed us to address these challenges. From October 12, 2021, to August 16, 2022, we conducted a global experience-sampling study in more than 40 countries as part of the "Coping with Corona" project (CoCo)[11]. Over a 4-week experience-sampling data collection period, participants reported their well-being in four randomly timed brief assessments per day, allowing us to track their momentary well-being. Additionally, participants completed online surveys on personality and sociodemographic variables before

✉ e-mail: julian.scharbert@uni-muenster.de

and after the experience-sampling period. These data offer a unique opportunity to trace well-being levels surrounding the outbreak of war in Ukraine over time and across countries.

In our preregistered analyses (osf.io/3uqtf), we focused on European countries because the outbreak of war has had the most direct consequences and was likely to be monitored most closely by the public in these countries[12]. Furthermore, to focus on the period during which the most important developments (to date) in the Russo-Ukrainian war took place, we targeted a two-month time window around the Russian invasion of Ukraine. The final European sample of $N = 1,341$ participants (44,894 experience-sampling reports) is illustrated in Fig. 1. Detailed, country-specific descriptive statistics are presented in Supplementary Table 1. All analyses were extended to and compared with all non-European countries in our sample as well as with broader and narrower time frames. In addition, we conducted several supplementary and robustness analyses (e.g., to investigate whether the results generalize to societal well-being and to control for country effects), the results of which are presented in the supplement.

Based on these analyses, we report an acute decline in global well-being levels on the day of the Russian invasion that affected individuals across nations and was not statistically significantly associated with their age, gender, social status, political orientation, or personality. Furthermore, we demonstrate that recovery in well-being over the weeks following the outbreak of war was slow and associated with an individual's personality, with individuals low in trait Stability showing close to no recovery effects, on average. Lastly, we show that well-being was particularly low on days when the war was more salient on social media.

## Results
### Well-being development surrounding the outbreak of war
To determine how individuals' well-being changed over the weeks before the Russians invaded Ukraine, on the day of the invasion, and in the weeks that followed, we adopted statistical procedures established in the literature on the psychological effect of personal life events[13,14]. That is, we fit eight multilevel models to the data, each representing different potential trajectories of well-being over time (Fig. 2). We then

compared the model fits to identify the model that best approximated the change in well-being around the outbreak of the war. The coefficients of all models are presented in the supplement (Supplementary Table 2). The best-fitting model was Model 2d, which is characterized by independent linear trends leading up to and following the Russian invasion of Ukraine and a sudden baseline level change on the day of the invasion.

The upper left quarter of Table 1 displays the fixed-effects coefficients of Model 2d. In addition, Fig. 3 displays the fixed and random slopes of Model 2d in our sample. The solid blue line illustrates the change in mean well-being as predicted by the model. The linear slope of well-being leading up to the Russian invasion was close to zero and not statistically significant ($b = 0.005$, 95% CI [−0.097, 0.107], $p = 0.923$). Thus, we found no credible evidence that participants experienced positive or negative anticipation effects in the weeks before the outbreak of the war, on average. On the day of the invasion, there was a sudden and significant decline in well-being ($b = −0.200$, 95% CI [−0.278, −0.122], $p < 0.001$), indicating that participants experienced an abrupt worsening of their well-being on that day, on average. In the following weeks, the linear slope was small but positive ($b = 0.089$, 95% CI [−0.006, 0.184], $p = 0.066$). Even though this increase was nonsignificant at $p < 0.05$, the effect size of the regression coefficient suggested that it took individuals approximately two months after the invasion to recover their initial well-being levels, on average. While this interpretation needs to be confirmed in future studies, it is in line with supplementary analyses considering narrower and broader time frames, in which the post-event slope was significant and positive (Supplementary Tables 11 and 13). Regarding country differences, we found some evidence of mean-level differences between countries and country-specific recovery effects that we discuss below (Supplementary Tables 5 and 7). Supplementary analyses on the data from our global sample yielded comparable results (Supplementary Table 9) and revealed that participants from European countries had significantly lower well-being compared with participants from non-European countries during the period surrounding the outbreak of the war ($b = −0.245$, 95% CI [−0.329, −0.162], $p < 0.001$).

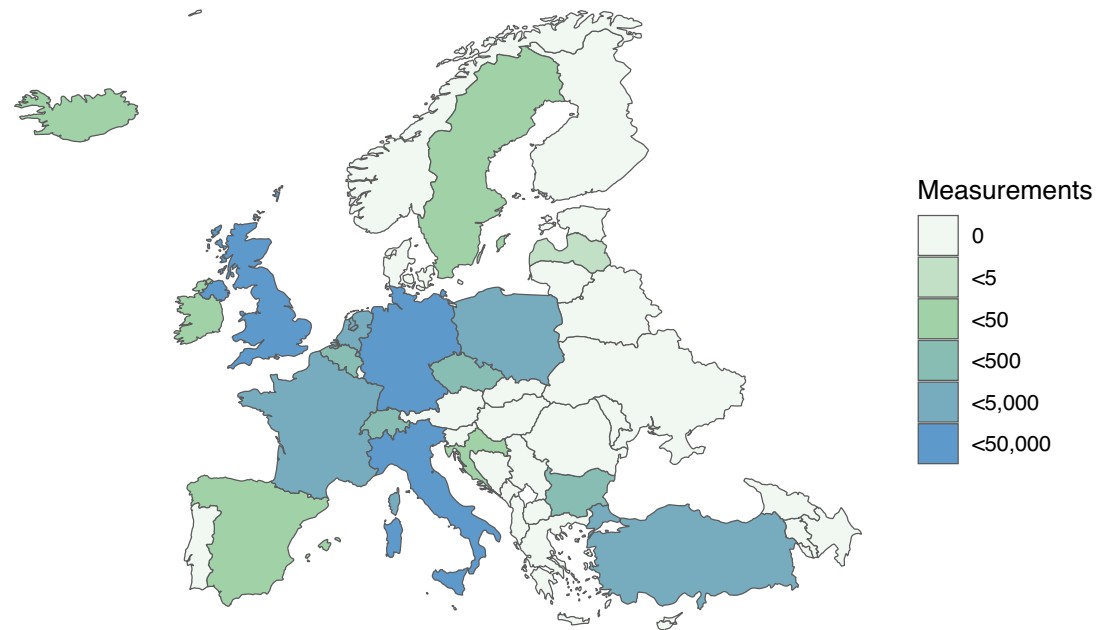

**Fig. 1 | Number of experience-sampling measurements in European countries between January 24 and March 27, 2022.** The number of measurements per country is indicated by the respective colour.

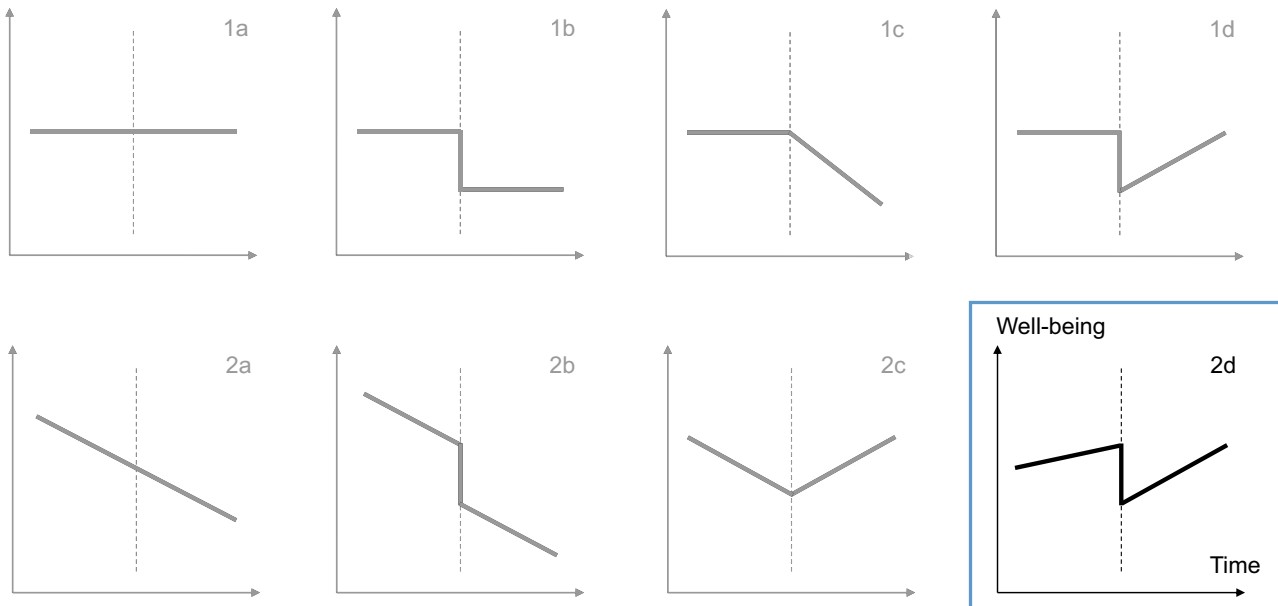

**Fig. 2 | Illustration of the eight models representing different theoretically possible trajectories of well-being over time.** The x-axes represent time; the y-axes represent mean daily well-being. The vertical dashed lines represent the day of the Russian invasion. The statistical operationalization of each model is presented in Table 3. Model 2d is highlighted because it fit the data best and was thus used for interpretation.

**Table 1 | Predicting well-being by time variables, personality, and the salience of the war**

|  | Individual well-being | | | | | | | |
|---|---|---|---|---|---|---|---|---|
| **Predictors** | **b** | **95%-CI** | **t** | **p** | **b** | **95%-CI** | **t** | **p** |
| **(Intercept)** | −0.090 | −0.156; −0.023 | −2.644 | 0.008 | −0.090 | −0.154; −0.025 | −2.714 | 0.007 |
| Level | −0.200 | −0.278; −0.122 | −5.040 | <0.001 | −0.182 | −0.260; −0.103 | −4.525 | <0.001 |
| Pre-event | 0.005 | −0.097; 0.107 | 0.096 | 0.923 | 0.041 | −0.061; 0.143 | 0.781 | 0.435 |
| Post-event | 0.089 | −0.006; 0.184 | 1.837 | 0.066 | 0.121 | 0.029; 0.212 | 2.571 | 0.010 |
| Stability | – | – | – | – | 0.245 | 0.180; 0.311 | 7.338 | <0.001 |
| Level * Stability | – | – | – | – | −0.027 | −0.102; 0.048 | −0.698 | 0.485 |
| Pre-event * Stability | – | – | – | – | −0.043 | −0.139; 0.054 | −0.867 | 0.386 |
| Post-event * Stability | – | – | – | – | 0.161 | 0.066; 0.255 | 3.320 | 0.001 |
| **(Intercept)** | −0.004 | −0.044; 0.035 | −0.216 | 0.829 | −0.008 | −0.048; 0.032 | −0.377 | 0.706 |
| Tweets | −0.070 | −0.096; −0.044 | −5.298 | <0.001 | – | – | – | – |
| Tweets (WS) | – | – | – | – | −0.065 | −0.092; −0.037 | −4.559 | <0.001 |
| Tweets (BS) | – | – | – | – | −0.116 | −0.207; −0.024 | −2.475 | 0.013 |

$N = 1341$. Displays fixed-effect coefficients of multilevel models. All statistical tests were two-sided. Degrees of freedom were >10,000 for all statistical tests.
*WS* within-subjects component, *BS* between-subjects component, *b* unstandardized regression weight, *95%-CI* 95% confidence interval around the estimate.

### Differences in reactions to the war

The thin gray lines in Fig. 3 represent the well-being trajectories of 100 randomly selected participants as predicted by Model 2d. Participants differed considerably in their individual slopes, especially in the weeks following the Russian invasion. To partly explain these individual differences in people's reactions to the outbreak of the war, we included the personality meta-trait Stability as a predictor of the individual slopes in Model 2d. Stability is a broad personality trait that reflects differences between people in how agreeable, conscientious, and emotionally stable they are[15,16]. Stability has been both theoretically[17] and empirically[18] linked to various aspects of well-being. Therefore, individual differences in Stability may help explain different reactions to the war. In supplementary analyses, we considered additional personality traits as well (i.e., the meta-trait Plasticity and the Big Five domains, Supplementary Tables 17 and 18). In addition to personality,

we investigated several sociodemographic variables (age, gender, subjective social status, political orientation) as potential predictors of the individual changes in well-being in Model 2d (Supplementary Table 3).

The two dashed lines in Fig. 3 represent the predicted changes in well-being in individuals whose Stability is one standard deviation above (upper line) or below (lower line) the sample mean. In addition, the results of Model 2d including Stability as a moderator are presented in the upper right quadrant of Table 1. Individuals high in Stability had an overall higher well-being compared with individuals low in Stability ($b = 0.245$, 95% CI [0.180, 0.311], $p < 0.001$). Stability was not statistically significantly associated with changes in well-being in the weeks leading up to and on the day of the Russian invasion ($b = −0.043$, 95% CI [−0.139, 0.054], $p = 0.386$ and $b = −0.027$, 95% CI [−0.102, 0.048], $p = 0.485$, respectively). However, it was significantly

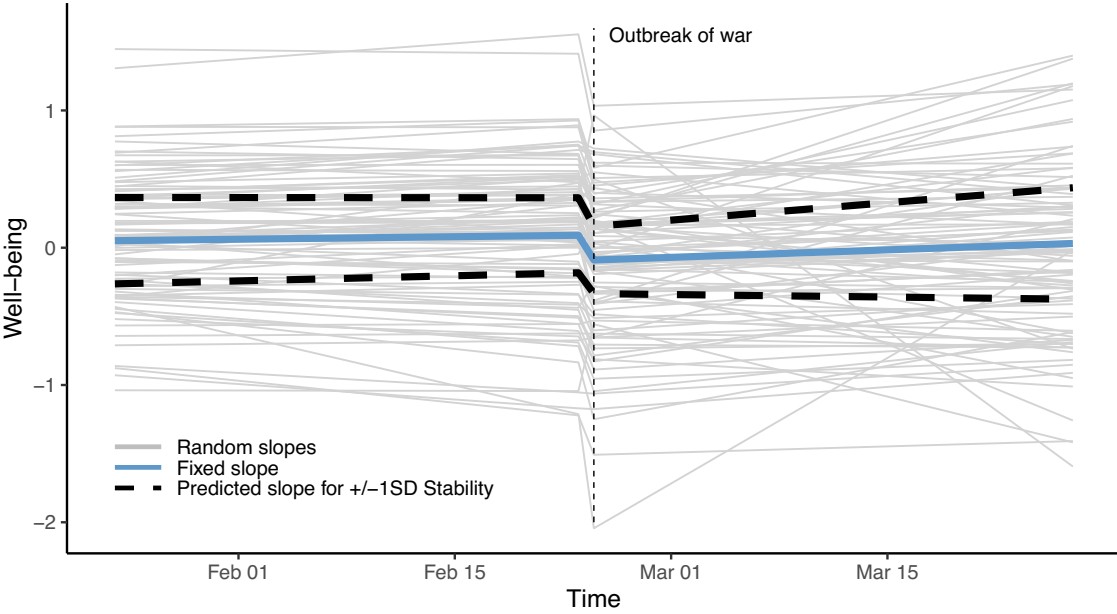

**Fig. 3 | Slopes of individual and mean well-being over time as predicted by Model 2d.** The solid blue line illustrates the predicted mean well-being trajectories. The thin gray lines represent the predicted well-being trajectories of 100 randomly chosen participants. The two dashed lines represent the predicted well-being trajectories of individuals with Stability scores of one standard deviation above (upper line; +1 SD) and below the mean (lower line, -1SD).

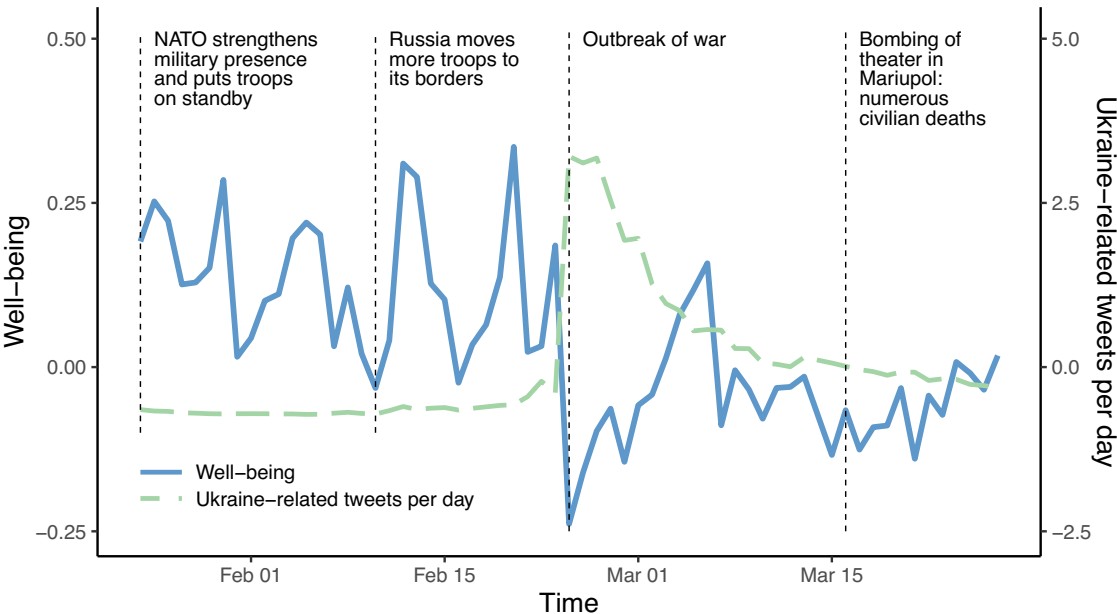

**Fig. 4 | Major events of the Russo-Ukrainian war and corresponding well-being levels and Ukraine-related tweets.** The solid blue line illustrates the mean daily well-being scores across participants (standardized across state measurements) and corresponds to the scale on the left. The dashed green line illustrates the number of Ukraine-related tweets each day (standardized across days) and corresponds to the scale on the right. The vertical dashed lines mark a selection of psychologically relevant events during the investigation period.

associated with recovery in well-being in the weeks after the invasion ($b = 0.161$, 95% CI [0.066, 0.255], $p = 0.001$). That is, while individuals high in Stability tended to recover comparatively quickly from the decline in well-being, individuals low in Stability tended to show close to no recovery effects in the first month after the invasion. We found some indication of country-specific effects of Stability on recovery, which we discuss below (Supplementary Tables 5 and 8). Compared with the sociodemographic variables, personality was more strongly

associated with well-being, and it was the only predictor that significantly explained variability in changes in well-being after the Russian invasion (Supplementary Table 3).

### Effect of the salience of the war

Lastly, we investigated whether daily well-being corresponded to the daily salience of the war. One potential path through which the outbreak of war may have influenced well-being is in the extent to which

individuals were confronted with the war and thereby started ruminating about the humanitarian catastrophes or the dangerous implications for themselves. To investigate this possibility, we retrieved the daily number of tweets that contained the keyword "Ukraine" worldwide and used this as a sample-independent indicator of the daily salience of the war on social media. Figure 4 depicts the mean daily well-being score in our sample and the number of Ukraine-related tweets each day worldwide. Both indicators showed related developments over the weeks that corresponded to the development of the war. For example, the decline in well-being on the day of the Russian invasion coincided with an increase in Ukraine-related tweets that slowly subsided over the following weeks.

In line with this visual observation, multilevel models (see the lower quadrants of Table 1) revealed that the number of Ukraine-related tweets were significantly negatively associated with daily well-being ($b = -0.070$, 95% CI [$-0.096$, $-0.044$], $p < 0.001$). We found a significant association with the between-subjects component (i.e., the average salience per person across all days on which they participated; $b = -0.116$, 95% CI [$-0.207$, $-0.024$], $p = 0.013$) and the within-subjects component of the salience (i.e., the daily deviation between salience on that day and the corresponding person's average salience; $b = -0.065$, 95% CI [$-0.092$, $-0.037$], $p < 0.001$). This finding indicates that individuals who participated on days with more Ukraine-related tweets had lower average well-being and that participants had lower average well-being on the days in their sampling period with more Ukraine-related tweets. Comparable associations were found for lagged associations (i.e., when predicting well-being from the number of Ukraine-related tweets from the previous day, Supplementary Table 15). We found no credible evidence that these effects were associated with individuals' personalities and sociodemographic characteristics (Supplementary Table 4).

## Discussion

Here, we used international, longitudinal experience-sampling data to track well-being levels across countries during the weeks surrounding the outbreak of the Russo-Ukrainian war. Multilevel analyses revealed that the Russian invasion of Ukraine was associated with an acute decline in participants' well-being in our sample. While we found no credible evidence that this initial decline in well-being was related to individuals' personality traits or their sociodemographic characteristics, there were systematic differences in how quickly individuals' well-being recovered from the initial decline. Specifically, the well-being of individuals high in the personality meta-trait Stability recovered more quickly than that of individuals low in Stability.

The fact that we found statistically significant associations between personality and recovery in well-being after the war but no credible evidence for associations between personality and the amplitude of the initial decline on the day of the Russian invasion might be explained by the high situational strength:[19] Waking up to smartphone notifications saying that the biggest country in the world had invaded a European country might have shocked people in similar ways, independent of their personalities or sociodemographic attributes. In the weeks that followed, when the initial shock had subsided, the situation had a potentially less uniform effect on individuals, such that differences in threat sensitivity, tendency to ruminate, and other traits covered by Stability might have led to differences in participants' propensities to experience a quick recovery in their well-being.

In general, individuals who participated on days when the war was especially salient on social media reported lower average well-being, and individuals reported lower average well-being on days when the war was more salient on social media. There is most likely a reciprocal relationship between daily well-being and salience, such that individuals worldwide posted more war-related tweets on days when they were particularly distressed by it, and individuals became more distressed by being exposed to war-related content on social media and elsewhere.

Over the weeks following the Russian invasion, when the war did not escalate to include further countries and its salience on social media decreased, participants tended to experience a recovery in their well-being, even though this recovery was slow and associated with participants' personality traits. This finding is in line with set-point models of well-being[9,20], which assert that good and bad events temporarily affect happiness, but that people quickly adapt back to their initial baseline level of well-being (a phenomenon that psychologists have referred to as the hedonic treadmill[21]). While this phenomenon was initially described as universal to human experience, more recent work has demonstrated that well-being set points can change under some conditions and that individuals differ in their adaptation to events[20,22]. Our results provide further support for these revisions to the hedonic treadmill model, as we found a normative adaption in well-being levels to the outbreak of war, on average, but substantial individual differences in such recovery effects that were related to personality traits.

Because the decline in well-being was only temporary for most participants, one might ask whether this reaction should be interpreted as a sign of dysfunctional distress or whether it could also be regarded as a natural or even adaptive response to the outbreak of war. For instance, such negative emotions might be one of the driving forces behind the overwhelming support for Ukraine (e.g., as expressed in donations or public peace demonstrations). Accordingly, it could be argued that trying to mitigate the negative effects on well-being with psychological interventions might reduce the support for Ukraine. Thus, to better understand the adaptiveness of the changes in well-being in response to the war, we conducted post-hoc (i.e., non-preregistered) supplementary analyses relating participants' well-being levels to various self-indicated Ukraine-related behaviors, emotions, and evaluations, as well as to different coping styles (Supplementary Table 21). In these exploratory analyses, we found that lowered well-being levels were mainly related to dysfunctional psychological indicators in our sample (e.g., worries about one's psychological health, anticipation of a third world war, avoidant coping strategies). We also found some associations between lower well-being and prosocial indicators such as stronger empathy with the affected people in the war zone, but we found no credible evidence that these translated to an increased solidarity with Ukraine (e.g., expressed through donations or participation in public protests). However, these post-hoc analyses are exploratory in nature. Future studies should examine more thoroughly how much lowered well-being in response to major societal crises includes maladaptive (e.g., overwhelming distress, worsening of clinical symptoms such as panic attacks, functional limitations such as disrupted sleep) and adaptive components (e.g., an expression of empathy and values such as peace and human rights).

Given that lowered well-being as a response to major societal crises is likely associated with dysfunctional stress at least for some individuals, it is warranted to consider ways to support individuals in coping with such crises. Rather than applying large-scale, expensive interventions for the whole population (which seems unrealistic given the many challenges arising after an event like the outbreak of war), institutions could focus on mental health campaigns that can be mobilized in a cost-effective and quick manner and that target those individuals who need them. For example, while increasing clinical capacities to supply individual psychotherapy to everyone in need will most likely not be possible, ad-hoc group therapy sessions, telephone helpline support, or web-based clinical interventions are relatively inexpensive yet still effective alternatives that could be rolled out in a timely manner[23–26]. Another option would be to promote and inform about self-administered interventions that affected individuals could apply when they feel particularly distressed. For example, expressive

writing[27] is a self-directed exercise in which one writes down all thoughts and feelings of currently stressful events. This method, which takes no more than 20–30 min, has been demonstrated to increase well-being and decrease anxiety, stress, and depressive symptoms[28,29]. However, we must note that we have not tested these direct implications for policy or clinical practice here, and the interventions mentioned are raised only as potential future implications. In addition, when administering or promoting such psychological interventions, it will always be important to consider the specific contexts and characteristics of the individuals, as well as the severity of their psychopathological symptoms. In particular, individuals who suffer from severe psychopathology following a major societal crisis (such as post-traumatic stress disorder) may require more intensive interventions such as individual psychotherapy. Indiscriminately applying solutions developed for more transient stress symptoms in these more severe cases might turn out to be ineffective or even detrimental to fostering psychological health[30].

Even though we cannot make strong direct causal inferences based on the data considered here, the outbreak of the war offers the most parsimonious explanation for the acute dip in well-being on the day of the Russian invasion and the associations found with the salience of the war on social media. Still, some limitations of our study should be noted. First, while our dataset includes many countries worldwide, participants are not equally distributed across these countries, with some countries (e.g., Italy, Germany) being overly represented and other countries contributing only a few participants. This imbalance makes it difficult to derive conclusions about the effects in single countries and the degree to which our results generalize to all countries in the world. Also, one needs to consider the unequal distribution of our data across time within these countries, as the number of daily reports peaked before the outbreak of war in some countries (e.g., Germany, UK) but during the weeks after the event in others (e.g., France, Italy). Robustness analyses accounting for mean-level differences between countries yielded no credible evidence that these differences affect the results in general. However, in exploratory supplementary analyses (not preregistered), we found some evidence for country-specific well-being trajectories. For instance, we found stronger recovery effects that were less strongly related to Stability when excluding Italy or weaker recovery effects that were more strongly related to Stability when excluding Turkey from the analyses. These findings suggest that the psychological effects of the outbreak of war in Ukraine might have differed between countries (e.g., due to their proximity to Ukraine or prior existing political tensions), but these differences are difficult to disentangle with our data. Future studies might combine separate datasets across countries collected during the weeks of the Russian invasion to circumvent the problem of unbalanced data distribution across countries and time. Second, our data were skewed towards more female, educated, and younger participants. This limits their representativeness for the general population, and future studies should investigate whether the effects of the outbreak of war change when including a more diverse and representative sample. Lastly, our sample did not include Ukrainian or Russian individuals for whom the psychological implications of the war have most likely been much larger. For example, Cheung and colleagues[31] found that the war in Syria had a devastating effect on the Syrian population's well-being with a magnitude of 1.1 SD. Even though it might be difficult to get reliable data from Ukraine and Russia regarding the mental health and well-being of their people, researchers should strive to uncover the psychological implications of the outbreak of war for these key populations.

Despite these limitations, we found evidence that the well-being of the European population temporarily declined after the outbreak of war in Ukraine. How large was the effect size of the decline? Compared with effect sizes found for other tragic life events, such as bereavement or disability (drops of around 0.6 to 0.7 SD)[9,32], we found a rather small effect size (0.2 SD). However, the previous studies investigated narrow subgroups of the population (e.g., disabled or bereaved people) and comparatively personal rather than collective life events. Considering that we investigated the decline in well-being in a broad sample following a global crisis, the effect size we found is considerably larger than in other studies of population-level global events, such as the 2020 COVID-19 lockdowns (no significant effects)[33] or the 2011 Fukushima disaster (drop in life satisfaction of around 0.12 SD for residents in the affected areas)[34]. As such, an important avenue for future research will be to identify relevant characteristics of life events that cause changes in well-being and to identify relevant characteristics of individuals who react resiliently to these events[35].

Since the outbreak of the war in Ukraine, governments, corporations, and public institutions have been hard-pressed to cope with its humanitarian, economic, and ecological consequences[3,8,36]. However, the potentially more covert psychological dimension of the outbreak of war should also not be neglected. In particular, individuals living in Europe with existing vulnerabilities (i.e., low levels of trait Stability) are likely to be struggling to cope with crises such as the outbreak of the war in Ukraine.

## Methods
Here, we describe all procedures and measures relevant to this manuscript. A codebook providing a full presentation of all applied procedures and measures in this project can be retrieved from the OSF (osf.io/8f3yu/). This study was preregistered on June 23, 2022, (osf.io/3uqtf) and represents the first publication based on data from this project. Our research complies with all relevant ethical regulations. The data collection for our project was approved by the institutional review board from the University of Münster (see below).

### Data collection procedure and ethical approval
Data were collected for this study as part of the "Coping with Corona" (CoCo) project, which comprises a global experience-sampling study with the goal of understanding individual differences in well-being during the COVID-19 pandemic[11]. Around the globe, participants took part in a 4-week study consisting of three phases: Upon registration, participants completed a large pre-survey that assessed personality traits and sociodemographic variables. Over the following four weeks, participants regularly completed up to four randomly timed short surveys throughout the day and an additional short survey at the end of each day. The surveys assessed their well-being, social interactions, attitudes, and more. After this 4-week experience-sampling period, participants completed a large post-survey that mostly consisted of the same items as the pre-survey.

All researchers involved in the project disseminated a link to the online survey through various channels including social media, local and digital blackboards, mailing lists, university classes, recruitment panels, and local press releases in their respective countries. As compensation, participants received personalized personality feedback throughout and after the data collection period. They could also take part in a raffle of 10,000€ (prizes ranged from 20€ to 2,500€), and we donated 1€ per participant to one of three charity organizations that participants could select. In Poland and the US, some participants additionally received direct monetary compensation, as they were recruited via research panels (Ariadna and TGM Research). The survey was conducted online via the open-source software formr version v0.18.3[37]. The data collection for the global project was hosted completely on a local server at the University of Münster, and no data point was at any moment stored outside Germany. The institutional review board from the University of Münster approved the complete international study (2020-54-MB). All participants provided informed consent.

## Data inclusion

This study represents a post-hoc analysis of data that had already been collected, so no statistical method was used a priori to predetermine sample size. All data inclusion procedures and decisions described here were preregistered on the Open Science Framework (osf.io/3uqtf) before the analyses were conducted. First, we restricted our data set to the period of one month (31 days) before and after the day of the Russian invasion (i.e., 31 days + 1 day + 31 days = 63 days; January 24 to March 27, 2022). Because we considered a symmetrical time frame around the outbreak of war, the linear developments before and after the event could be compared directly. We additionally conducted supplementary analyses in which we focused on a narrower (7 days before and after the Russian invasion) and a broader period (January 1 to April 19, 2022, the broadest possible symmetrical time frame around the Russian invasion excluding the Christmas holidays) to investigate whether the effect sizes found varied depending on the time frame. Second, all participants from non-European countries were excluded from our main analyses. These participants were included in supplementary analyses with the global sample. Third, we applied three additional data exclusion criteria and excluded a) participants who indicated on the last page of the post-questionnaire that they had not answered the questions in the survey conscientiously, b) participants who completed the pre-questionnaire too quickly (who took less than two seconds per item on average), and c) participants who provided state data on less than two days (this was minimally required to have variance on the change parameters).

These procedures resulted in $N = 1341$ participants (44,894 experience-sampling reports) in the main analyses, $N = 341$ (5604 experience-sampling reports) in the analyses with a shorter time frame, $N = 1915$ (76,716 experience-sampling reports) in the analyses with a longer time frame, and $N = 1735$ (54,851 experience-sampling reports) in the analyses with the global data. Of the 1341 participants included in the main analyses, 1079 were women, 252 were men, and 10 specified another gender. The mean age was 25.7 (min = 18, max = 91). Country-specific demographic statistics are presented in the supplement (Supplementary Table 1).

On March 14, 2022, there was a technical error that resulted in the submission of only a few assessments on that day. Hence, to avoid displaying a misrepresentative well-being score for this day, Fig. 4 does not include any data from this day. However, the assessments from this day were included in all other analyses.

## Personality measures

The Big Five traits were measured with the 60-item Big Five Inventory-2[38]. Items were answered on a 5-point scale ranging from 1 (disagree strongly) to 5 (agree strongly). Stability and Plasticity were calculated by aggregating individuals' Agreeableness, Conscientiousness, and reversed Neuroticism scores (Stability) and their Extraversion and Openness scores (Plasticity). We considered only the trait data assessed in the pre-survey, because not all participants provided data in the post-survey.

## Sociodemographic variables

For the sociodemographic variables, we assessed age, gender, subjective social status, and political orientation. Participants reported their age directly. For gender, participants could specify female, male, or another gender. As only ten participants specified another gender, we investigated gender effects by differentiating between female and male participants only. We assessed subjective social status with one item similar to the MacArthur Scale of Subjective Social Status:[39] Participants were asked to imagine a 10-step ladder representing the social positions of the population in their country and to indicate where they would place themselves and their family on this ladder ranging from 1 (at the bottom) to 10 (at the top). For political orientation, participants were asked to indicate how they generally view their political orientation on a scale ranging from 1 (extreme left) to 10 (extreme right).

## Well-being measures

Well-being was operationalized as the aggregate of positive affect and reverse-scored negative affect (also referred to as affective well-being[40,41]). Positive and negative affect were assessed in the randomly timed experience-sampling surveys throughout the day with three items each ("I felt happy", "...enthusiastic", and "...relaxed" for positive affect; "I felt angry", "...afraid", and "...sad" for negative affect). Items were answered on a 6-point scale ranging from 1 (do not agree at all) to 6 (agree completely). As we expected that developments in the war would fluctuate more between days than within days, we aggregated the state-level data to the daily level for each participant. Daily well-being was calculated as the mean of the daily positive affect and reversed daily negative affect scores. In the main analyses, this procedure resulted in 17,990 daily well-being scores.

## Salience of the war in Ukraine

The salience of the war in Ukraine was operationalized as the global number of daily tweets that contained the keyword "Ukraine". We retrieved this tweet count via the Twitter Researcher API v2. Days were defined within the UTC time zone.

## Measures for supplementary analyses

For supplementary and post-hoc analyses described below, we assessed societal well-being and various Ukraine-related behaviors, emotions, evaluations, and coping styles. Societal well-being was assessed daily in the evening questionnaire and operationalized as the combination of positive evaluations of, low threat perceptions towards, and high perceived similarity with people living in one's country. The positivity of other evaluations was assessed with the item "How positive or negative have you felt towards people in your country in general today?", answered on a 10-point scale ranging from 1 (very negative) to 10 (very positive). Threat perceptions were assessed with the item "How threatened have you felt by people in your country in general today?", answered on a 10-point scale ranging from 1 (not at all threatened) to 10 (extremely threatened). The perceived similarity was assessed with the item "How similar have you felt to people in your country in general today?", answered on a 10-point scale ranging from 1 (not at all similar) to 10 (extremely similar). Daily societal well-being was calculated by aggregating the daily positivity of other evaluations, reversed daily threat perceptions, and daily perceived similarity.

The Ukraine-related items were included in most languages (excluding Portuguese, Polish, Turkish, and Thai) after the outbreak of the war and were introduced into the pre and post surveys between March 16 and March 23, 2022. For participants who provided answers in both the pre and post survey, we only used the answers from the pre survey because they were closer in time to the outbreak of the war. The items assessing Ukraine-related behaviors, emotions, and evaluations were items created by the authors that are presented in Supplementary Table 21. Most items were answered on a 6-point scale ranging from 1 (Strongly disagree) to 6 (Strongly agree). The item "How much news do you consume in comparison to before the war in Ukraine?" was answered with a slider ranging from 1 (clearly less) to 10 (clearly more) and the item "I consider the political proceedings of my country concerning the war in Ukraine…" was answered with a slider ranging from 1 (very wrong) to 10 (very right). Ukraine-related coping styles were assessed with an adapted version of the 28-item Brief COPE[42]. This questionnaire assesses 14 coping strategies that can be aggregated into the three broader coping styles problem-focused, emotion-focused, and avoidant coping. The original instructions were changed so that participants explicitly referred to the Ukraine war when answering the items. In addition, some of the items were changed so that they made sense given the instructions (e.g., "I've been

concentrating my efforts on doing something about the situation I'm in" was changed to "I've been concentrating my efforts on doing something about the situation in Ukraine."). Items were answered on a 4-point scale ranging from 1 (Not at all) to 4 (A lot).

## Statistical analyses

All analyses were conducted using the software *R* version 4.2.1[43] and the packages lme4[44] and brms[45]. All statistical tests were two-sided with a significance threshold of $p < 0.05$. All quasi-continuous trait-level variables (personality, age, subjective social status, political orientation, Ukraine-related variables) were *z*-standardized across participants. The numbers of Ukraine-related tweets were *z*-standardized across days. The well-being scores were *z*-standardized across all measurement points across all days. Note that there is no established consensus on how one should obtain standardized coefficients in multilevel models[46] and our predictor variables were *z*-standardized in different ways (or not standardized in case of the change parameters described below). Thus, we refrain from referencing conventional benchmarks for effect sizes and refer to our model coefficients as unstandardized regression weights. However, as well-being scores were *z*-standardized across all measurement points across all days, model coefficients can be interpreted as changes in well-being on that scale.

To identify how well-being developed over time, we fit eight multilevel models to the data in which daily scores (Level 1) were nested within individuals (Level 2). Each model contained a different combination of change parameters on Level 1 resulting in different changes in well-being over the weeks. The statistical operationalization of these parameters is illustrated in Table 2. Each parameter can be interpreted in the following way: (1) when the time variable was used as a predictor of individuals' daily well-being, its effect represented linear changes in well-being over the weeks independent of the Russian invasion; (2) the respective effect of the level variable represented baseline level changes in well-being before versus after the Russian invasion; (3) the effect of the pre-event variable represented linear changes in the weeks leading up to the Russian invasion, indicating negative or positive anticipation effects; (4) the effect of the post-event variable represented linear changes in the weeks following the Russian invasion, indicating recovery or worsening effects over time.

The time as well as the pre- and post-event variables were rescaled so that they ranged from −1 to 1 (time), −1 to 0 (pre-event), and 0 to 1 (post-event) over the 63 days investigated in the main analyses. Therefore, the regression weights indicated average changes in well-being from the first day of the period investigated in the main analyses to the day of the invasion (pre-event) or average changes from the day of the invasion to the last day of the period investigated in the main analyses (post-event). Accordingly, these variables ranged from −0.194 to 0.194 in the analyses with a shorter and from −1.742 to 1.742 in the analyses with a longer time frame. Note that the values of the change parameters were the same for every person on any given date, but there was nonetheless between-person variance in the values of the change parameters aggregated across all days for individuals because

participants could enter the survey at any time (resulting in individually shifted participation periods).

We focused on eight multilevel models that result from different combinations of these parameters and that each represented different trajectories of well-being over time. All models were specified as random-intercept-random-slope models because model comparisons showed significant differences between the fit of the models with and without random slopes (i.e., random-intercept-constant-slope models). We compared the empirical evidence for these models simultaneously using the Akaike Information Criterion (AIC) because this allowed us to compare all models, including those that were not nested within each other. Table 3 presents all models with their conceptual meaning, Level-1 equations, and a graphical illustration. The Level-2 equations are analogous for all models; for example, the full equation of Model 2d for time point i nested in person j is as follows, with the other models being defined accordingly:

Level 1:

$$WB_{ij} \sim \beta'_{0j} + \beta'_{1j} preLin_i + \beta'_{2j} postLin_i + \beta'_{3j} level_i + \varepsilon_{ij}$$

Level 2:

$$1. \beta'_{0j} = \beta_0 + \beta^*_{0j}$$

$$2. \beta'_{1j} = \beta_1 + \beta^*_{1j}$$

$$3. \beta'_{2j} = \beta_2 + \beta^*_{2j}$$

$$4. \beta'_{3j} = \beta_3 + \beta^*_{3j}$$

The best-fitting model was Model 2d (AIC = 42,781). Model 2d had slightly more evidence in the data than Model 1d (AIC = 42,791) even though the fixed effect of the pre-event variable was not statistically significant in Model 2d. This difference between the models is likely a combined effect of the freely estimated fixed effect of the pre-event variable (which was nonsignificant but nonzero) and the freely estimated random effects of the pre-event variable (i.e., differences in anticipation effects between persons) in Model 2d. Model 2d was than extended by including Stability and the sociodemographic variables as Level-2 predictors of the random intercepts and random slopes in separate models, respectively. To investigate the influence of the salience of the war on daily well-being, we specified multilevel models in which the number of daily Ukraine-related tweets was used as a Level-1 predictor of daily well-being. We additionally split the salience indicator into a between-subjects component (the average salience per person across all days on which they participated) and a within-subjects component (the daily deviation between salience on that day and the respective person's average salience) and included both as predictors of daily well-being in the model. In the analyses investigating individual differences in the effects, we included Stability and the sociodemographic variables as Level-2 predictors of both the random intercepts and random slopes.

We deviated from the preregistration by using maximum likelihood (ML) estimation instead of restricted ML (REML) because REML would not have allowed us to compare model fit using the AIC. As we faced model convergence issues for some of the models with the frequentist approach, we calculated all models using a Bayesian estimator as a robustness check. Here, we used the default, uninformative priors in the brms package[47] and estimated the parameters based on four chains with 20,000 iterations each and a delta value of 0.99. These models converged and yielded results that were very similar to those from the frequentist analyses, replicating all reported significant

## Table 2 | Illustration of the change parameters around the Russian invasion

| Date | 24.01. | ... | 22.02. | 23.02. | 24.02. | 25.02. | 26.02. | ... | 27.03. |
|---|---|---|---|---|---|---|---|---|---|
| Time | −1 | ... | −0.064 | −0.032 | 0 | 0.032 | 0.064 | ... | 1 |
| Level | −1 | ... | −1 | −1 | 0 | 0 | 0 | ... | 0 |
| Pre-event | −1 | ... | −0.064 | −0.032 | 0 | 0 | 0 | ... | 0 |
| Post-event | 0 | ... | 0 | 0 | 0 | 0.032 | 0.064 | ... | 1 |

**Table 3 | Illustration of the interpretation, statistical definition, and illustration of the models**

| Model | Interpretation | Level-1 model for time point i nested in person j | Illustration of one possible trajectory |
|---|---|---|---|
| 1a | No change at all | $WB_{ij} \sim \beta'_{0j} + \varepsilon_{ij}$ |  |
| 1b | Baseline level change after Russian invasion | $WB_{ij} \sim \beta'_{0j} + \beta'_{1j} level_i + \varepsilon_{ij}$ |  |
| 1c | Gradual change after Russian invasion | $WB_{ij} \sim \beta'_{0j} + \beta'_{1j} postLin_i + \varepsilon_{ij}$ |  |
| 1d | Baseline level change after Russian invasion, subsequent gradual change | $WB_{ij} \sim \beta'_{0j} + \beta'_{1j} postLin_i + \beta'_{2j} level_i + \varepsilon_{ij}$ |  |
| 2a | Gradual change independent of Russian invasion | $WB_{ij} \sim \beta'_{0j} + \beta'_{1j} time_i + \varepsilon_{ij}$ |  |
| 2b | Gradual change and baseline level change after Russian invasion | $WB_{ij} \sim \beta'_{0j} + \beta'_{1j} time_i + \beta'_{2j} level_i + \varepsilon_{ij}$ |  |
| 2c | Gradual change leading up to Russian invasion, independent subsequent gradual change | $WB_{ij} \sim \beta'_{0j} + \beta'_{1j} preLin_i + \beta'_{2j} postLin_i + \varepsilon_{ij}$ |  |
| 2d | Gradual change leading up to Russian invasion, baseline level change after Russian invasion, subsequent gradual change | $WB_{ij} \sim \beta'_{0j} + \beta'_{1j} preLin_i + \beta'_{2j} postLin_i + \beta'_{3j} level_i + \varepsilon_{ij}$ |  |

"preLin" and "postLin" correspond to the linear pre-event and post-event slope.

effects and yielding almost identical effect sizes (Supplementary Tables 22 to 40).

**Supplementary analyses**

We conducted several supplementary analyses. First, because our data stemmed from different countries, we ran supplementary analyses for all main analyses in which we additionally included countries with at least 10 participants as dummy-coded variables as predictors of random intercepts in the models (the remaining countries were used as the reference category, Supplementary Tables 5 and 6). As additional robustness analyses (not preregistered) on potential differences between countries, we excluded one of these countries at a time from the analyses to identify changes in well-being over time (and Stability predicting these changes, Supplementary Tables 7 and 8). Second, we investigated whether the effect sizes found differed between European countries and the rest of the world. To this end, we conducted the main analyses with the complete, global sample and included a dummy-coded variable for European versus non-European countries as a predictor of the random intercepts (Supplementary Tables 9 and 10). Third, we limited and extended our data set to the narrower and broader time frames, respectively, to investigate whether the effect sizes found varied by the investigated period (Supplementary Tables 11 to 14). Fourth, we calculated models with the salience of the war in Ukraine predicting daily well-being on the consecutive day to check for lagged associations (Supplementary Table 15). Fifth, we ran Model 2d including Stability as a predictor with the sub-facets of well-being (i.e., positive and negative affect). Analogously, we calculated the models with the respective Big Five domains instead, as well as with the second meta-trait, Plasticity, and its respective Big Five domains (Supplementary Tables 16 and 17). Sixth, we ran all main analyses with societal instead of individual well-being as an outcome to investigate the specificity of the results found for individual well-being (Supplementary Tables 18 to 20). Lastly, we conducted post-hoc supplementary analyses in which we used the Ukraine-related variables as Level-2

predictors of individuals' well-being in random-intercept multilevel models (Supplementary Table 21).

## Reporting summary

Further information on research design is available in the Nature Portfolio Reporting Summary linked to this article.

## Data availability

The raw and processed data that support the findings of this study are publicly available on the OSF (https://doi.org/10.17605/OSF.IO/8F3YU). The data collection was part of the "Coping with Corona" project[11], which includes additional variables that we did not consider in this manuscript.

## Code availability

All analysis scripts can be obtained from the OSF (https://doi.org/10.17605/OSF.IO/8F3YU).

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

## Acknowledgements

This research was supported by funding provided to Mitja D. Back, Markus Bühner, and Maarten van Zalk by the German Research Foundation under grant numbers BA 3731/11-1, BU 1414/3-1, and ZA 1147/1-1, and by funding provided to Clemens Stachl by the Swiss National Foundation under grant number 215303. The authors wish to acknowledge the following individuals for contributing to the research: Carolin Albrecht, Cosima Heinen, Ricarda Luther, and Anna Schnickmann.

## Author contributions

J.S., L.K., T.R., S.S., J.t.H., K.U., M.v.Z., M.B., and M.D.B. developed the concept for the 'Coping with Corona' project. J.S., T.R., S.S., J.t.H, M.v.Z., M.B., and M.D.B. were responsible for the technical implementation and international roll-out of the data collection. J.S. developed the concept of the research question and analytical strategy with critical feedback from S.H. and M.D.B.; J.S. was responsible for the data analysis, and S.H., L.K., and M.D.B. provided critical feedback on it. J.S. drafted the manuscript. All remaining authors were involved in the international data collection and/or critical revisions of the manuscript. All authors approved the final version of the manuscript.

## Funding

## Competing interests

The authors declare no competing interests.

## Additional information

Julian Scharbert [1] ✉, Sarah Humberg[1], Lara Kroencke [1], Thomas Reiter [2], Sophia Sakel[2], Julian ter Horst [3], Katharina Utesch [1], Samuel D. Gosling [4,5], Gabriella Harari [6], Sandra C. Matz[7], Ramona Schoedel[2], Clemens Stachl [8], Natalia M. A. Aguilar[9], Dayana Amante [10], Sibele D. Aquino [11], Franco Bastias [12], Alireza Bornamanesh[13], Chloe Bracegirdle[14], Luís A. M. Campos[11,15], Bruno Chauvin [16], Nicoleen Coetzee[17], Anna Dorfman[18], Monika dos Santos[19], Rita W. El-Haddad[20], Malgorzata Fajkowska[21], Asli Göncü-Köse [22], Augusto Gnisci[23], Stavros Hadjisolomou [20], William W. Hale III[24], Maayan Katzir [25], Lili Khechuashvili[26], Alexander Kirchner-Häusler [27], Patrick F. Kotzur[28], Sarah Kritzler[29], Jackson G. Lu [30], Gustavo D. S. Machado[31], Khatuna Martskvishvili [26], Francesca Mottola[23], Martin Obschonka [32], Stefania Paolini [28], Marco Perugini[33], Odile Rohmer[16], Yasser Saeedian[34], Ida Sergi[23], Maor Shani[3], Ewa Skimina[35], Luke D. Smillie [5], Sanaz Talaifar [36], Thomas Talhelm[37], Tülüce Tokat [38], Ana Torres [39], Claudio V. Torres [40], Jasper Van Assche [41,42], Liuqing Wei[43], Aslı Yalçın [22], Maarten van Zalk[3], Markus Bühner[2] & Mitja D. Back[1,44]

[1]Department of Psychology, University of Münster, Münster, Germany. [2]Department of Psychology, University of Munich, Munich, Germany. [3]Department of Psychology, Osnabrück University, Osnabrück, Germany. [4]Department of Psychology, University of Texas at Austin, Austin, USA. [5]School of Psychological Sciences, The University of Melbourne, Melbourne, Australia. [6]Department of Communication, Stanford University, Stanford, USA. [7]Business School,

Columbia University, New York, USA. [8]Institute of Behavioral Science and Technology, University of St. Gallen, St. Gallen, Switzerland. [9]Faculty of Veterinary Sciences, National University of the Northeast, Corrientes, Argentina. [10]Research Institute in Basic and Applied Psychology, Catholic University of Cuyo, San Juan, Argentina. [11]Department of Psychology, Pontifical Catholic University of Rio de Janeiro, Rio de Janeiro, Brazil. [12]Cluster of Excellence "The Politics of Inequality", University of Konstanz, Konstanz, Germany. [13]Psychiatry Department, Isfahan University of Medical Sciences, Isfahan, Iran. [14]Nuffield College, University of Oxford, Oxford, England. [15]Department of Psychology, Catholic University of Petrópolis, Petrópolis, Brazil. [16]Department of Psychology, University of Strasbourg, Strasbourg, France. [17]Department of Psychology, University of Pretoria, Pretoria, South Africa. [18]Department of Psychology, Bar Ilan University, Ramat Gan, Israel. [19]Department of Psychology, University of South Africa, Pretoria, South Africa. [20]Department of Social and Behavioral Sciences, American University of Kuwait, Safat, Kuwait. [21]Institute of Psychology, Polish Academy of Sciences, Warsaw, Poland. [22]Department of Psychology, Çankaya University, Ankara, Turkey. [23]Department of Psychology, University of Campania Luigi Vanvitelli, Caserta, Italy. [24]Department of Youth and Family, Utrecht University, Utrecht, Netherlands. [25]Conflict Resolution, Management, and Negotiation Graduate Program, Bar Ilan University, Ramat Gan, Israel. [26]Department of Psychology, Ivane Javakhishvili Tbilisi State University, Tbilisi, Georgia. [27]School of Psychology, University of Sussex, Brighton, England. [28]Department of Psychology, Durham University, Durham, England. [29]Department of Psychology, Ruhr University Bochum, Bochum, Germany. [30]Sloan School of Management, Massachusetts Institute of Technology, Cambridge, USA. [31]Department of Psychology, Federal University of Santa Catarina, Florianópolis, Brazil. [32]Amsterdam Business School, University of Amsterdam, Amsterdam, the Netherlands. [33]Department of Psychology, University of Milan-Bicocca, Milan, Italy. [34]Institute for Physical Activity and Nutrition, Deakin University, Burwood, Australia. [35]Institute of Psychology, SWPS University, Warsaw, Poland. [36]Department of Management & Entrepreneurship, Imperial College London, London, England. [37]Booth School of Business, The University of Chicago, Chicago, USA. [38]Human Sciences Department, Verona University, Verona, Italy. [39]Department of Psychology, Federal University of Paraíba, João Pessoa, Brazil. [40]Department of Basic Psychological Processes, University of Brasilia, Brasilia, Brazil. [41]Department of Developmental, Personality and Social Psychology, Ghent University, Ghent, Belgium. [42]Center for Social and Cultural Psychology (CESCUP), Université Libre de Bruxelles, Brussels, Belgium. [43]Department of Psychology, Hubei University, Wuhan, China. [44]Joint Institute for Individualisation in a Changing Environment (JICE), University of Münster and Bielefeld University, Münster, Germany. ✉e-mail: julian.scharbert@uni-muenster.de

