## [Peer Review File · Nature Communications]

Reviewers' Comments:

Reviewer #1:

Remarks to the Author:

The current manuscript examines how the outbreak of the Russo-Ukrainian War shapes well-being using ESM data from different countries. The data are unique, and the analytical plan is well-thought-out. I think the manuscript can be strengthened if the authors take the following concerns into consideration.

1. The assumption that greater well-being is good: I acknowledge that this critique may be less empirical but more philosophical. The current manuscript is written with an assumption that greater well-being is good / worse well-being is bad. A clear example is that the authors – in their concluding paragraph – noted that 'We should strive to offer affected individuals help in preserving their well-being'; Lines 250-251.' I have major reservation over this assumption, and this hidden assumption could be costly. Given the nature of the war, negative emotion is arguably the adaptive response and likely underlies the motivation behind the overwhelming support for Ukraine (e.g., in the forms of hosting refugees, donating supplies, and supporting policies around providing humanitarian and military aids to Ukraine). To intervene on it given the context of the war could mean less support for Ukraine.

As another example, the analyses on social media also imply that exposure to social media is bad (because it leads to worsened well-being; lines 206-208). However, increasing one's awareness of the war through social media could again be an adaptive response. E.g., the anger one feels about the injustice and the sadness of witnessing the brutality of war may be needed for one's continued support for Ukraine.

We can also consider a counterfactual – based on the current paper's reasoning, finding an improvement in well-being (i.e., people becoming happier) following the war may be interpreted as a good thing.

To be constructive, I think the authors' framing would be more effective if they provide evidence that participants indeed experienced 'distress' (e.g., in the form of functional limitations or clinical diagnosis). Alternatively, if the authors provide evidence that the lowered well-being is linked to behavioral outcomes related to supporting Ukraine, that would also allow psychological researchers to critically rethink whether interventions are always needed. With the current evidence, it is difficult to know how the results should be interpreted and acted on, and the authors' preferred interpretation could be potentially insensitive in a wartime context.

2. The interpretation of recovery: The authors wrote in line 98-100, "In the following weeks, the linear slope was positive but nonsignificant ($b = 0.089 [-0.006, 0.184]$, $p = .066$), indicating that participants' well being did not significantly recover during the first month after the invasion, on average." I understand the authors' interpretation (based on a 0.05 cut-off). My interpretation based on the regression coefficient of 0.089 is that on average, the initial 0.2SD drop in well-being is likely to recover fully in ~ 2 months ($0.089 * 2 \sim 0.2$). That doesn't strike me as a long-term drop. The interpretation that the well-being drop is temporary is also supported by other supplementary analyses (e.g., those discussed in line 102-103 on different time frames). Taken together, the initial drop followed a recovery seems to be more aligned with my interpretation (concern #1) that such drop in well-being does not necessarily reflect distress and could be an adaptive response to an unusual circumstance.

The relative short-lived negative impact with a modest effect size also seem to stand in sharp contrast with the overall framing on the paper. E.g., Lines 247-248, "However, neglecting the potentially more covert effect of the outbreak of war on mental well-being could also have severe consequences." I disagree with characterizing this change in well-being as severe.

The time frame and overall recovery also calls the authors' recommendations on mental health campaigns and personality-tailored well-being interventions into question. The authors can strengthen the policy implications by citing mental health campaigns that can be mobilized in a cost-effective manner within 2 months. Moreover, given the relatively quick recovery of mental well-being, directing time and resources toward it may take scarce resources away from those in

needs.

3. Countries and samples: The abstract promises a study on 17 countries, which is technically true, but there are 10 countries where there are fewer than 10 participants. Italy (n = 659) alone composed of almost half the total participants (N = 1341). Based on descriptive statistics presented in Supplementary Table 1, the sample is not representative, but this is neither acknowledged in the abstract nor the limitation section. And the title, abstract, and conclusion (and throughout the manuscript) seem to imply that the findings are broadly applicable/generalizable. I hope the authors can transparently acknowledge these limitations.

Reviewer #2:

Remarks to the Author:

Review "How the Outbreak of War in Ukraine Impaired Psychological Well Being Across Nations"

In the present study, the authors investigated the impact of the Russian invasion of Ukraine on the well-being of individuals living in other European countries. By using a quite large subset of an even larger global research project employing experience sampling data, the authors observed a sharp drop in well-being on the day of the outbreak of the war followed by a steady recovery until the end of the observation period. The well-being trajectory has been found to be predicted by inter-individual differences in Stability and to be furthermore associated with the salience of the war on Twitter. Overall, there is a lot to admire about the present work, including the large data set, the thorough analytical approach accompanied by extensive and informative supplemental analyses, the creative and elegant operationalization of media salience, and the authors' commitment to open science by pre-registering their analysis and by releasing the relevant data and analysis scripts. To be honest, I found the present paper already quite impressive in its current form; as a consequence, my comments are much more suggestions and requests than severely critical points.

1. I very much appreciate the extensive supplement with the detailed additional analyses. In particular the country-level analyses were very informative and a sensible extension to the main analyses. While the results are very well illustrated in the main text, I found it sometimes difficult to make sense of the effects purely based on the supplementary tables. Thus, I would very much appreciate if the authors could add some figures illustrating, for example, country-level effects in the Supplement.

2. Somewhat related to the previous point, the authors mention in the main text that they have found country-level differences in the well-being trajectories and announced to discuss them later. However, I think this discussion is not very detailed and basically only mentions that these differences occurred. It would be interesting to discuss these effects in slightly more detail.

3. If I understood the data collection procedure correctly, participants could join the study anytime. As a consequence, the experience sampling schedule might differ between people, as the ESM part started after the pre-survey was completed. If this is indeed the case, I wondered how the authors dealt with these individually "shifted" data. If this is not the case or if participants were selected based on their "entry" into the ESM phase, a more explicit description of this aspect may be helpful.

4. I understand that the present study is exploratory in nature. However, embedding the findings a bit more into the literature, for example on set-point models of well-being may be worthwhile.

Reviewer #3:

Remarks to the Author:

This paper elucidates the impact of the war in Ukraine on individuals' mental well-being outside Ukraine using an experience-sampling method. It is noteworthy that the authors conducted various kinds of supplemental analyses as well as main analyses to support their claim that the

war that took place in Ukraine devastated mental well-being of Europeans and non-Europeans that participated in this study. Overall, I found this manuscript informative and interesting, but I have some comments meant to strengthen the final product.

1. The authors recruited participants by disseminating a link to the survey online. As a result, participants were relatively young, and females were oversampled. How can we be sure that the participants were representative samples and the results will generalize to other populations? The authors should mention this limitation in the discussion.

2. The descriptions in the results section (Impact of war on well-being) sounded like the model 1d might fit the data as well (no anticipatory effect, but sudden decrease in well-being on the day of the invasion, and marginal recovery after the invasion). But it was not the case. Was the authors' choice of the models solely based on AIC? If the authors use other criteria (e.g., chi-square values) to test model fit, would the results be the same?

Relatedly, what is the difference between model 1d and model 2d? Doesn't the linear slope of well-being before the Russian invasion in model 2d have to be a significant change in well-being before the outbreak of the war?

3. Whereas the authors provided descriptive statistics of European samples, they did not provide those of non-European samples. It would be nice to have more information about non-European countries (e.g., countries participants belonged to, descriptive statistics) as well in supplemental materials.

REVIEWER 1:

R1: The current manuscript examines how the outbreak of the Russo-Ukrainian War shapes well-being using ESM data from different countries. The data are unique, and the analytical plan is well-thought-out. I think the manuscript can be strengthened if the authors take the following concerns into consideration.

Thank you for your positive comments. We have addressed all your valuable points as described below and agree that considering them has strengthened the manuscript.

R1.1: The assumption that greater well-being is good: I acknowledge that this critique may be less empirical but more philosophical. The current manuscript is written with an assumption that greater well-being is good / worse well-being is bad. A clear example is that the authors – in their concluding paragraph – noted that ‘We should strive to offer affected individuals help in preserving their well-being’; Lines 250-251.’ I have major reservation over this assumption, and this hidden assumption could be costly. Given the nature of the war, negative emotion is arguably the adaptive response and likely underlies the motivation behind the overwhelming support for Ukraine (e.g., in the forms of hosting refugees, donating supplies, and supporting policies around providing humanitarian and military aids to Ukraine). To intervene on it given the context of the war could mean less support for Ukraine. As another example, the analyses on social media also imply that exposure to social media is bad (because it leads to worsened well-being; lines 206-208). However, increasing one’s awareness of the war through social media could again be an adaptive response. E.g., the anger one feels about the injustice and the sadness of witnessing the brutality of war may be needed for one’s continued support for Ukraine. We can also consider a counterfactual – based on the current paper’s reasoning, finding an improvement in well-being (i.e., people becoming happier) following the war may be interpreted as a good thing.

To be constructive, I think the authors’ framing would be more effective if they provide evidence that participants indeed experienced ‘distress’ (e.g., in the form of functional limitations or clinical diagnosis). Alternatively, if the authors provide evidence that the lowered well-being is linked to behavioral outcomes related to supporting Ukraine, that would also allow psychological researchers to critically rethink whether interventions are always needed. With the current evidence, it is difficult to know how the results should be interpreted and acted on, and the authors’ preferred interpretation could be potentially insensitive in a wartime context.

Thank you for raising this important point and constructively suggesting ways to address it. We agree that highlighting the potentially adaptive and maladaptive components of well-being changes in response to a crisis is an important nuance that was missing in our prior version of the manuscript. Hence, we have included a paragraph in the discussion section to raise awareness of this point:

“Because the decline in well-being was only temporary for most participants, one might ask whether this reaction should be interpreted as a sign of dysfunctional distress or whether it could also be regarded as a natural or even adaptive response to the outbreak of war. For instance, such negative emotions might be one of the driving forces behind the overwhelming support for Ukraine (e.g., as expressed in donations or public peace demonstrations). Accordingly, trying to mitigate the negative effects on well-being with psychological interventions might reduce the support for Ukraine. Taking this argument to the extreme, finding completely neutral reactions to the war (or even increases in well-being) could be considered insensitive (or even outrageous).”

While the question of whether lowered well-being in the weeks surrounding the outbreak of war in Ukraine was predominantly adaptive or maladaptive is complex and deserves to be investigated thoroughly in future studies, we have tried to approach this question in additional post-hoc analyses. Concretely, we have related participants' well-being levels to various self-indicated Ukraine-related behaviors, emotions, and evaluations, as well as to different coping styles. In short, our analyses showed that lower well-being levels were related to rather maladaptive consequences, such as stronger worries and more avoidant coping strategies. We have added the following paragraph in the method section describing the assessment of these Ukraine-related variables:

“The Ukraine-related items were included in most languages (excluding Portuguese, Polish, Turkish, and Thai) after the outbreak of the war and were introduced into the pre and post surveys between March 16 and March 23, 2022. For participants who provided answers in both the pre and post survey, we only used the answers from the pre survey because they were closer in time to the outbreak of the war. The items assessing Ukraine-related behaviors, emotions, and evaluations were self-created items that are presented in Supplementary Table 21. Most items were answered on a 6-point scale ranging from 1 (*Strongly disagree*) to 6 (*Strongly agree*). The item “How much news do you consume in comparison to before the war in Ukraine?” was answered with a slider ranging from 1 (*clearly less*) to 10 (*clearly more*) and the item “I consider the political proceedings of my country concerning the war in Ukraine...” was answered with a slider ranging from 1 (*very wrong*) to 10 (*very right*). Ukraine-related coping styles were assessed with an adapted version of the 28-item Brief COPE⁴⁴. This questionnaire assesses 14 coping strategies that can be aggregated into the three broader coping styles problem-focused, emotion-focused, and avoidant coping. The original instructions were changed so that participants explicitly referred to the Ukraine war when answering the items. In addition, some of the items were changed so that they made sense given the instructions (e.g., “I've been concentrating my efforts on doing something about the situation I'm in” was changed to “I've been concentrating my efforts on doing something about the situation in Ukraine.”). Items were answered on a 4-point scale ranging from 1 (*Not at all*) to 4 (*A lot*).”

In our post-hoc supplementary analyses, we have used these Ukraine-related variables as level-2 predictors of individuals' well-being in random-intercept multilevel models. The results of these analyses together with the concrete item wordings are presented in Supplementary Table 21, which we also include below. We have summarized these results in the following paragraph in the discussion section:

“In post-hoc supplementary analyses relating participants' well-being levels to various self-indicated Ukraine-related behaviors, emotions, and evaluations, as well as to different coping styles (Supplementary Table 21), we found that lowered well-being levels were mainly related to dysfunctional psychological indicators in our sample (e.g., worries about one's psychological health, anticipation of a third world war, avoidant coping strategies). While we also found some associations between lower well-being and prosocial indicators such as stronger empathy with the affected people in the war zone, we found no credible evidence that these translated to an increased solidarity with Ukraine (e.g., expressed through donations or participation in public protests). However, these post-hoc analyses are exploratory in nature. Future studies should examine more thoroughly how much lowered well-being in response to major societal crises includes maladaptive (e.g., overwhelming distress, worsening of clinical symptoms such as panic attacks, functional limitations such as disrupted sleep) and adaptive components (e.g., an expression of empathy and values such as peace and human rights).”

As written above, we acknowledge that these analyses are exploratory in nature and provide no strong evidence for the (non-)adaptiveness of the psychological impact of the outbreak of the war.

Nonetheless, they suggest that lowered well-being was associated with dysfunctional stress for at least some individuals, making it reasonable to discuss potential ways of helping these affected individuals in coping with the crisis. Additionally, the interventions that we discuss below (see our response to R1.2) do not need to result in individuals simply being unaffected or indifferent to the horrors of the war (e.g., as one might expect if we advertised prescribing anti-anxiety medication to large parts of the population). Rather, we have attempted to suggest interventions that help individuals find ways to productively deal with their emotions and the situation (e.g., through reflective exercises and opening up about their distress), potentially increasing the likelihood that individuals choose proactive coping strategies (e.g., trying to help individuals in need or donating to war-related charities).

We hope you agree that the changes to our manuscript as inspired by your comment have added nuance to the framing of changes in well-being, have helped us to suggest more appropriate and concrete low-cost intervention strategies, and have increased the likelihood that policymakers adopt political strategies that contribute to helping vulnerable individuals help in coping with this and future crises.

Predictors		Individual well-being			
		b	95%-CI	t	p
Ukraine-related behavior 1	"How much news do you consume in comparison to before the war in Ukraine?" (higher values = more news)	0.033	-0.024; 0.089	1.135	0.257
Ukraine-related behavior 2	"I show my solidarity with Ukraine (for example through donations, participation in protests or public positioning)."	0.040	-0.017; 0.096	1.382	0.167
Ukraine-related behavior 3	"I purposefully reduce my news consumption concerning the war in Ukraine."	0.044	-0.012; 0.101	1.535	0.125
Ukraine-related behavior 4	"I prepare myself for further escalation of the conflict by keeping vital goods in stock (e.g., water, toilet paper, canned products, or petrol)."	-0.120	-0.176; -0.064	-4.235	<0.001
Ukraine-related emotion 1	"I am worried about my physical well-being due to the war in Ukraine."	-0.119	-0.175; -0.063	-4.185	<0.001
Ukraine-related emotion 2	"I am worried about the physical well-being of my family due to the war in Ukraine."	-0.109	-0.165; -0.053	-3.84	<0.001
Ukraine-related emotion 3	"I am worried about my psychological well-being due to the war in Ukraine."	-0.142	-0.197; -0.086	-5.014	<0.001
Ukraine-related emotion 4	"I am worried when I consume news about the war in Ukraine."	-0.096	-0.153; -0.040	-3.362	0.001
Ukraine-related emotion 5	"I am worried that the war will spread to other countries."	-0.094	-0.150; -0.038	-3.304	0.001
Ukraine-related emotion 6	"I am worried about the economic situation in my country."	-0.100	-0.156; -0.043	-3.469	0.001
Ukraine-related emotion 7	"I strongly feel for the affected people in the war zone."	-0.074	-0.130; -0.017	-2.545	0.011
Ukraine-related evaluation 1	"I assume that due to the war in Ukraine the third world war will come."	-0.105	-0.161; -0.049	-3.696	<0.001
Ukraine-related evaluation 2	"I consider the political proceedings of my country concerning the war in Ukraine ..." (higher values = stronger approval)	0.107	0.051; 0.163	3.723	<0.001
Ukraine-related evaluation 3	"I consider an increase in the defense budget in my country reasonable."	0.036	-0.021; 0.092	1.235	0.217
Ukraine-related evaluation 4	"I consider reporting of the public media in my country concerning the war in Ukraine factual."	0.053	-0.003; 0.110	1.847	0.065
Brief COPE: problem-focused coping	active coping, use of informational support, planning, and positive reframing	0.098	0.042; 0.155	3.435	0.001
Brief COPE: emotion-focused coping	venting, use of emotional support, humor, acceptance, self-blame, and religion	0.075	0.018; 0.131	2.588	0.010
Brief COPE: avoidant coping	self-distraction, denial, substance use, and behavioral disengagement	-0.095	-0.151; -0.038	-3.292	0.001

R1.2: The interpretation of recovery: The authors wrote in line 98-100, “In the following weeks, the linear slope was positive but nonsignificant ($b = 0.089 [-0.006, 0.184]$, $p = .066$), indicating that participants’ well being did not significantly recover during the first month after the invasion, on average.” I understand the authors’ interpretation (based on a 0.05 cut-off). My interpretation based on the regression coefficient of 0.089 is that on average, the initial 0.2SD drop in well-being is likely to recover fully in ~2 months ($0.089 * 2 \sim 0.2$). That doesn’t strike me as a long-term drop. The interpretation that the well-being drop is temporary is also supported by other supplementary analyses (e.g., those discussed in line 102-103 on different time frames). Taken together, the initial drop followed a recovery seems to be more aligned with my interpretation (concern #1) that such drop in well-being does not necessarily reflect distress and could be an adaptive response to an unusual circumstance.

The relative short-lived negative impact with a modest effect size also seem to stand in sharp contrast with the overall framing on the paper. E.g., Lines 247-248, “However, neglecting the potentially more covert effect of the outbreak of war on mental well-being could also have severe consequences.” I disagree with characterizing this change in well-being as severe.

The time frame and overall recovery also calls the authors’ recommendations on mental health campaigns and personality-tailored well-being interventions into question. The authors can strengthen the policy implications by citing mental health campaigns that can be mobilized in a cost-effective manner within 2 months. Moreover, given the relatively quick recovery of mental well-being, directing time and resources toward it may take scarce resources away from those in needs.

Thank you for your comments. We agree that assisting readers with interpreting the regression coefficients and relating the magnitude of the recovery effect to the initial decline is helpful. Therefore, we have added the following paragraph in the results section:

“In the following weeks, the linear slope was small but positive ($b = 0.089 [-0.006, 0.184]$, $p = .066$). Even though this increase was nonsignificant at $p < .05$, the effect size of the regression coefficient suggested that it took individuals approximately two months after the invasion to recover their initial well-being levels, on average. While this interpretation needs to be confirmed in future studies, it is in line with supplementary analyses considering narrower and broader time frames, in which the post-event slope was significant and positive.”

To address your disagreement with characterizing the change in well-being as “severe”, we have changed the wording throughout the manuscript to avoid making judgments about the magnitude of the effect and rather point out the fact that a negative effect exists. For example, in the abstract, we now write about a “detrimental” rather than a “harmful” psychological impact. As another example, in the conclusion, we write that “the potentially more covert effect of the outbreak of war on mental well-being should also not be neglected” rather than suggesting that neglecting the consequences could have “severe consequences”.

Nonetheless, our analyses show that participants differed considerably in how quickly they recovered in their well-being levels and that there might be particularly vulnerable individuals who mentally suffered from the outbreak of war for a prolonged time. In combination with the results discussed above regarding the maladaptive component of lowered well-being levels, we consider it important to think about potential ways to help these affected individuals. Still, we agree with your comment that this demands mental health campaigns that can be mobilized in a cost-effective and timely manner. We have attempted to do so with the following paragraph, which we have added to the discussion section:

“Given that lowered well-being as a response to major societal crises is likely associated with dysfunctional stress at least for some individuals, governments should consider ways to support individuals in coping with such crises. Rather than applying large-scale, expensive

interventions for the whole population (which seems unrealistic given the many challenges arising after an outbreak of war), governments could focus on mental health campaigns that can be mobilized in a cost-effective and quick manner and that target particularly those individuals who feel in need of them. For example, while increasing clinical capacities to supply individual psychotherapy to everyone in need will most likely not be possible, ad-hoc group therapy sessions, telephone helpline support, or web-based clinical interventions are relatively inexpensive but effective alternatives that could be rolled out in a timely manner²²⁻²⁵. Another option would be to promote and inform about self-administered interventions that affected individuals could apply when they feel particularly distressed. For example, *expressive writing*²⁶ is a self-directed exercise in which one writes down all thoughts and feelings of currently stressful events. This method, which takes no more than 20 to 30 minutes, has been demonstrated to increase well-being and decrease anxiety, stress, and depressive symptoms^{27,28}.”

Lastly, primarily to address comment R2.4 by Reviewer 2 but also related to your comment, we have discussed the fact that we did find normative recovery effects over ~2 months by relating this finding to set-point models of well-being in the discussion section:

“Over the weeks following the Russian invasion, when the war did not escalate to include further countries and its salience on social media decreased, participants tended to experience a recovery in their well-being, even though this recovery was slow and depended on participants’ personality traits. This finding is in line with set-point models of well-being^{9,19}, which assert that good and bad events temporarily affect happiness, but that people quickly adapt back to their initial baseline level of well-being (a phenomenon that psychologists have referred to as the *hedonic treadmill*²⁰). While this phenomenon was initially described as universal to human experience, more recent work has demonstrated that well-being set points can change under some conditions and that individuals differ in their adaptation to events^{19,21}. Our results provide further support for these revisions to the hedonic treadmill model, as we found a normative adaption in well-being levels to the outbreak of war, on average, but substantial individual differences in such recovery effects that were related to personality traits.”

R1.3: Countries and samples: The abstract promises a study on 17 countries, which is technically true, but there are 10 countries where there are fewer than 10 participants. Italy (n = 659) alone composed of almost half the total participants (N = 1341). Based on descriptive statistics presented in Supplementary Table 1, the sample is not representative, but this is neither acknowledged in the abstract nor the limitation section. And the title, abstract, and conclusion (and throughout the manuscript) seem to imply that the findings are broadly applicable/generalizable. I hope the authors can transparently acknowledge these limitations.

We agree that including the technically true number of countries from which our participants come may create false expectations. Therefore, we have excluded these numbers throughout the manuscript to avoid confusion and refer the reader directly to Supplementary Table 1 to obtain an accurate overview of the distribution of participants across countries.

Furthermore, we have emphasized this point in the limitation section:

“First, while our dataset includes many countries worldwide, participants are not equally distributed across these countries, with some countries (e.g., Italy, Germany) being overly represented and other countries contributing only a few participants. This imbalance makes it

difficult to derive conclusions about the effects in single countries and the degree to which our results generalize to all countries in the world.”

REVIEWER 2:

R2: In the present study, the authors investigated the impact of the Russian invasion of Ukraine on the well-being of individuals living in other European countries. By using a quite large subset of an even larger global research project employing experience sampling data, the authors observed a sharp drop in well-being on the day of the outbreak of the war followed by a steady recovery until the end of the observation period. The well-being trajectory has been found to be predicted by inter-individual differences in Stability and to be furthermore associated with the salience of the war on Twitter. Overall, there is a lot to admire about the present work, including the large data set, the thorough analytical approach accompanied by extensive and informative supplemental analyses, the creative and elegant operationalization of media salience, and the authors' commitment to open science by pre-registering their analysis and by releasing the relevant data and analysis scripts. To be honest, I found the present paper already quite impressive in its current form; as a consequence, my comments are much more suggestions and requests than severely critical points.

Thank you for your positive comments and the helpful suggestions.

R2.1: I very much appreciate the extensive supplement with the detailed additional analyses. In particular the country-level analyses were very informative and a sensible extension to the main analyses. While the results are very well illustrated in the main text, I found it sometimes difficult to make sense of the effects purely based on the supplementary tables. Thus, I would very much appreciate if the authors could add some figures illustrating, for example, country-level effects in the Supplement.

We discuss this comment together with the following one as the two are related.

R2.2: Somewhat related to the previous point, the authors mention in the main text that they have found country-level differences in the well-being trajectories and announced to discuss them later. However, I think this discussion is not very detailed and basically only mentions that these differences occurred. It would be interesting to discuss these effects in slightly more detail.

We agree that presenting detailed, country-level analyses of the effects would be very valuable and that illustrating country-specific trajectories in figures comparable to the figures in the main text would help in comparing countries directly. However, such figures would most likely be misleading given the unequal distribution of our data across countries and time. As we discussed in our initial submission, the sample sizes are not equally distributed over the weeks surrounding the outbreak of war, and some countries contributed considerably more participants to the overall sample in the weeks after the Russian invasion compared to the weeks before the invasion and vice versa. Accordingly, we think that our data are best suited to investigate trajectories on an international level by combining all data on any given day to obtain the most reliable estimates of our model parameters. The same applies to the presented figures, which are the most representative and reliable when including all data points from all countries.

Still, we agree that this point could be discussed in more detail, which is why we have edited the following paragraph in the limitation section in which we also discuss the country-specific effects in more detail and suggest ways in which future studies might obtain more detailed insights regarding differences between countries:

“First, while our dataset includes many countries worldwide, participants are not equally distributed across these countries, with some countries (e.g., Italy, Germany) being overly represented and other countries contributing only a few participants. This imbalance makes it difficult to derive conclusions about the effects in single countries and the degree to which our results generalize to all countries in the world. Also, one needs to consider the unequal distribution of our data across time within these countries, as the number of daily reports peaked before the outbreak of war in some countries (e.g., Germany, UK) but during the weeks after the event in others (e.g., France, Italy). Robustness analyses accounting for mean-level differences between countries indicated that these differences did not affect the results in general. However, in exploratory supplementary analyses, we found some evidence for country-specific well-being trajectories over the weeks. For instance, we found stronger recovery effects that were less strongly related to Stability when excluding Italy or weaker recovery effects that were more strongly related to Stability when excluding Turkey from the analyses. These findings suggest that the psychological effects of the outbreak of war in Ukraine might have differed between countries (e.g., due to their proximity to Ukraine or prior existing political tensions), but these differences are difficult to disentangle with our data. Future studies might combine separate datasets across countries collected during the weeks of the Russian invasion to circumvent the problem of unbalanced data distribution across countries and time.”

In addition, we have added references to the supplementary tables throughout the results section to make it easier for readers to navigate the supplement, understand the content of the supplementary tables, and find relevant information more quickly.

R2.3: If I understood the data collection procedure correctly, participants could join the study anytime. As a consequence, the experience sampling schedule might differ between people, as the ESM part started after the pre-survey was completed. If this is indeed the case, I wondered how the authors dealt with these individually "shifted" data. If this is not the case or if participants were selected based on their "entry" into the ESM phase, a more explicit description of this aspect may be helpful.

You understood correctly that participants could join the study anytime and this means that the 28-day participation periods were individually shifted. This was accounted for by the fact that the change parameters had the same value for every person on any given date, which is illustrated in Table 2 in the method section:

Date	24.01.	...	22.02.	23.02.	24.02.	25.02.	26.02.	...	27.03.
Time	-1	...	-0.064	-0.032	0	0.032	0.064	...	1
Level	-1	...	-1	-1	0	0	0	...	0
Pre-event	-1	...	-0.064	-0.032	0	0	0	...	0
Post-event	0	...	0	0	0	0.032	0.064	...	1

This means that two participants who started on different days had different values for their first day, second day, and so on. For example, imagine a hypothetical Participant 1 who started on January 24, 2022, and a hypothetical Participant 2 who started on February 22, 2022. The “pre-event” parameter would have a value of -1 on Day 1 for Participant 1, whereas it would have a value of -0.064 on Day 1

for Participant 2. This way, there is between-person variance in the values of the change parameters aggregated across all days for both participants.

We hope that this example illustrates how we dealt with the individually shifted data. We have included an additional explanation in the method section to clarify this point for readers as well:

“Note that the values of the change parameters were the same for every person on any given date, but there was nonetheless between-person variance in the values of the change parameters aggregated across all days for individuals because participants could enter the survey at any time (resulting in individually shifted participation periods).”

R2.4: I understand that the present study is exploratory in nature. However, embedding the findings a bit more into the literature, for example on set-point models of well-being may be worthwhile.

Thank you for the valuable suggestion to embed the findings more into the literature and the helpful reference to the set-point model of well-being as a relevant point of discussion. We have added a corresponding paragraph in the discussion section:

“Over the weeks following the Russian invasion, when the war did not escalate to include further countries and its salience on social media decreased, participants tended to experience a recovery in their well-being, even though this recovery was slow and depended on participants’ personality traits. This finding is in line with set-point models of well-being^{9,19}, which assert that good and bad events temporarily affect happiness, but that people quickly adapt back to their initial baseline level of well-being (a phenomenon that psychologists have referred to as the *hedonic treadmill*²⁰). While this phenomenon was initially described as universal to human experience, more recent work has demonstrated that well-being set points can change under some conditions and that individuals differ in their adaptation to events^{19,21}. Our results provide further support for these revisions to the hedonic treadmill model, as we found a normative adaption in well-being levels to the outbreak of war, on average, but substantial individual differences in such recovery effects that were related to personality traits.”

Also, addressing some comments of Reviewer 1, we have added the following two paragraphs to the discussion section to discuss our results in more detail:

“Because the decline in well-being was only temporary for most participants, one might ask whether this reaction should be interpreted as a sign of dysfunctional distress or whether it could also be regarded as a natural or even adaptive response to the outbreak of war. For instance, such negative emotions might be one of the driving forces behind the overwhelming support for Ukraine (e.g., as expressed in donations or public peace demonstrations). Accordingly, trying to mitigate the negative effects on well-being with psychological interventions might reduce the support for Ukraine. Taking this argument to the extreme, finding completely neutral reactions to the war (or even increases in well-being) could be considered insensitive (or even outrageous). In post-hoc supplementary analyses relating participants’ well-being levels to various self-indicated Ukraine-related behaviors, emotions, and evaluations, as well as to different coping styles (Supplementary Table 21), we found that lowered well-being levels were mainly related to dysfunctional psychological indicators in our sample (e.g., worries about one’s psychological health, anticipation of a third world war, avoidant coping strategies). While we also found some associations between lower well-being and prosocial indicators such as stronger empathy with the affected people in the war zone, we

found no credible evidence that these translated to an increased solidarity with Ukraine (e.g., expressed through donations or participation in public protests). However, these post-hoc analyses are exploratory in nature. Future studies should examine more thoroughly how much lowered well-being in response to major societal crises includes maladaptive (e.g., overwhelming distress, worsening of clinical symptoms such as panic attacks, functional limitations such as disrupted sleep) and adaptive components (e.g., an expression of empathy and values such as peace and human rights).

Given that lowered well-being as a response to major societal crises is likely associated with dysfunctional stress at least for some individuals, governments should consider ways to support individuals in coping with such crises. Rather than applying large-scale, expensive interventions for the whole population (which seems unrealistic given the many challenges arising after an event like the outbreak of war), governments could focus on mental health campaigns that can be mobilized in a cost-effective and quick manner and that target particularly those individuals who feel in need of them. For example, while increasing clinical capacities to supply individual psychotherapy to everyone in need will most likely not be possible, ad-hoc group therapy sessions, telephone helpline support, or web-based clinical interventions are relatively inexpensive but effective alternatives that could be rolled out in a timely manner²²⁻²⁵. Another option would be to promote and inform about self-administered interventions that affected individuals could apply when they feel particularly distressed. For example, *expressive writing*²⁶ is a self-directed exercise in which one writes down all thoughts and feelings of currently stressful events. This method, which takes no more than 20 to 30 minutes, has been demonstrated to increase well-being and decrease anxiety, stress, and depressive symptoms^{27,28}.

REVIEWER 3:

R3: This paper elucidates the impact of the war in Ukraine on individuals' mental well-being outside Ukraine using an experience-sampling method. It is noteworthy that the authors conducted various kinds of supplemental analyses as well as main analyses to support their claim that the war that took place in Ukraine devastated mental well-being of Europeans and non-Europeans that participated in this study. Overall, I found this manuscript informative and interesting, but I have some comments meant to strengthen the final product.

Thank you for your positive comments and the helpful suggestions.

R3.1: The authors recruited participants by disseminating a link to the survey online. As a result, participants were relatively young, and females were oversampled. How can we be sure that the participants were representative samples and the results will generalize to other populations? The authors should mention this limitation in the discussion.

We agree that the skewed demographic limits the representativeness of our sample and that this point should be discussed in some more detail in the limitation section. While it remains an open empirical question to which degree our results will generalize to other populations, we believe that there are good reasons to assume that the results will generalize and that increasing the heterogeneity in the sample might even uncover stronger effects of the outbreak of the war on mental well-being. Our main argument is that the restricted demographic in our sample is likely to reduce variability in personality as well as individual well-being, given that variables such as gender, age, and education tend to relate

to both. Including a more representative sample regarding age, gender, and education might potentially uncover even stronger systematic differences in well-being surrounding the outbreak of war.

Nonetheless, we acknowledge that a more representative sample is always desirable and have included an additional paragraph in the limitations section to discuss this point:

“Second, our data were skewed towards more female, educated, and younger participants. This limits their representativeness for the general population, and future studies should investigate whether the effects of the outbreak of war change when including a more diverse and representative sample. Such an increased heterogeneity in the sample might uncover even stronger effects of the outbreak of the war, given that the observed variability in both personality and well-being is likely to increase within a more diverse sample.”

R3.2: The descriptions in the results section (Impact of war on well-being) sounded like the model 1d might fit the data as well (no anticipatory effect, but sudden decrease in well-being on the day of the invasion, and marginal recovery after the invasion). But it was not the case. Was the authors' choice of the models solely based on AIC? If the authors use other criteria (e.g., chi-square values) to test model fit, would the results be the same?

Relatedly, what is the difference between model 1d and model 2d? Doesn't the linear slope of well-being before the Russian invasion in model 2d have to be a significant change in well-being before the outbreak of the war?

Thank you for pointing out that this finding might cause some confusion to the reader so that we can hopefully communicate this more clearly in our revised manuscript.

We based our choice of models on the AIC because not all models were nested and could be compared directly by testing for a significant difference in log likelihoods. For example, Model 2b is not nested in Model 2d because of the “time” parameter. Thus, the AIC is the better choice to evaluate model fit, because it allowed us to compare the empirical evidence for all eight models simultaneously.

Still, to address your point, we have conducted a post-hoc analysis in which we compared Model 1d with Model 2d directly (which is possible in this case because Model 1d is nested in Model 2d). This test confirmed our conclusion based on the AIC, as Model 2d fitted significantly better compared to Model 1d, $\chi^2(5) = 20.735, p < .001$.

This result makes sense despite the finding that the fixed effect of the pre-event variable was not significant in Model 2d because all models were specified as random-intercept-random-slope models. That is, the slightly better fit of Model 2d compared to Model 1d was likely a combined effect of the freely estimated fixed effect of the pre-event variable (which was nonsignificant but nonzero) and the freely estimated random effects of the pre-event variable (i.e., differences in anticipation effects between persons) in Model 2d.

To clarify this point to the readers, we have included additional explanations in the method section:

“All models were specified as random-intercept-random-slope models because model comparisons showed significant differences between the fit of the models with and without random slopes (i.e., random-intercept-constant-slope models). We compared the empirical evidence for these models simultaneously using the Akaike Information Criterion (AIC) because this allowed us to compare all models, including those that were not nested within each other.”

and

“The best-fitting model was Model 2d (AIC =42,781). Model 2d had slightly more evidence in the data than Model 1d (AIC = 42,791) even though the fixed effect of the pre-event variable was not significant in Model 2d. This difference between the models is likely a combined effect of the freely estimated fixed effect of the pre-event variable (which was nonsignificant but nonzero) and the freely estimated random effects of the pre-event variable (i.e., differences in anticipation effects between persons) in Model 2d.”

R3.3: Whereas the authors provided descriptive statistics of European samples, they did not provide those of non-European samples. It would be nice to have more information about non-European countries (e.g., countries participants belonged to, descriptive statistics) as well in supplemental materials.

Thank you for the helpful suggestion. We have exchanged Supplementary Table 1 with an extended version that includes the non-European countries as well.

Reviewers' Comments:

Reviewer #2:

Remarks to the Author:

Review "How the Outbreak of War in Ukraine Impaired Psychological Well Being Across Nations"

I already served as a reviewer for the initial submission of this manuscript. After going through the response letter and parts of the manuscript again, I thank the authors for their responsiveness. My concerns have been addressed satisfactorily and very diligently; I have no further comments. I wish the authors all the best for their current and future research and beyond.

Reviewer #4:

Remarks to the Author:

While I did not have the opportunity to review the first submission of this manuscript, I have reviewed the revision. In my opinion, the authors have adequately addressed the reviewers' comments. I have two points for consideration for the authors.

First, in response to Reviewer 1's comments regarding the interpretation of "wellbeing" in the context of war, the authors have re-conceptualized some of their findings. I agree with the authors' overall approach in relation to this but note in their revision that they state "Taking this argument to the extreme, finding completely neutral reactions to the war (or even increases in well-being) could be considered insensitive (or even outrageous)". While I agree with the spirit of what the authors are trying to achieve here, I wondered if this could be re-framed. In the context of scientific enquiry, the findings are the findings, irrespective of the context. Accordingly, it seems to me that a finding in itself cannot be insensitive or outrageous. Rather, care needs to be taken to interpret findings in the broader context.

Second, the authors refer to potential public health approaches to addressing mental health needs in the context of warfare. While I agree with the broad idea of providing low-cost pragmatic approaches, I think it is important to consider the evidence base when making these recommendations. For example, different approaches are likely to be warranted for individuals experiencing transient distress (which is common following adversity) compared to those who are showing signs of more severe psychopathology that may warrant intensive intervention. Approaches like Critical Incident Stress Debriefing have been demonstrated to be ineffective (or at worst, harmful) when indiscriminately applied in the aftermath of a traumatic event - these could potentially be conceptualized as ad-hoc group therapy sessions. I think this section could use some more consideration in terms of the specific interventions being recommended, ensuring that is specified who they would be recommended for, and that they are supported by research evidence.

REVIEWER 2:

R2: I already served as a reviewer for the initial submission of this manuscript. After going through the response letter and parts of the manuscript again, I thank the authors for their responsiveness. My concerns have been addressed satisfactorily and very diligently; I have no further comments. I wish the authors all the best for their current and future research and beyond.

We would like to thank Reviewer 2 for the positive evaluation and helpful suggestions during the review process.

REVIEWER 4:

R4: While I did not have the opportunity to review the first submission of this manuscript, I have reviewed the revision. In my opinion, the authors have adequately addressed the reviewers' comments. I have two points for consideration for the authors.

We would like to thank Reviewer 4 for reviewing our response to the original reviewers' suggestions. We address the two additionally suggested points below.

R4.1: First, in response to Reviewer 1's comments regarding the interpretation of "wellbeing" in the context of war, the authors have re-conceptualized some of their findings. I agree with the authors' overall approach in relation to this but note in their revision that they state "Taking this argument to the extreme, finding completely neutral reactions to the war (or even increases in well-being) could be considered insensitive (or even outrageous)". While I agree with the spirit of what the authors are trying to achieve here, I wondered if this could be re-framed. In the context of scientific enquiry, the findings are the findings, irrespective of the context. Accordingly, it seems to me that a finding in itself cannot be insensitive or outrageous. Rather, care needs to be taken to interpret findings in the broader context.

Thank you for rightfully commenting that a finding in itself cannot be insensitive or outrageous. We agree with this criticism of our previous framing. Therefore, we have removed this sentence and slightly edited this section, which now reads:

“[...] Accordingly, trying to mitigate the negative effects on well-being with psychological interventions might reduce the support for Ukraine. Thus, to better understand the adaptiveness of the changes in well-being in response to the war, we conducted post-hoc (i.e., non-preregistered) supplementary analyses relating participants' well-being levels to various self-indicated Ukraine-related behaviors, emotions, and evaluations, as well as to different coping styles (Supplementary Table 21). In these exploratory analyses, we found that lowered well-being levels were mainly related to dysfunctional psychological indicators in our sample (e.g., worries about one's psychological health, anticipation of a third world war, avoidant coping strategies).”

R4.2: Second, the authors refer to potential public health approaches to addressing mental health needs in the context of warfare. While I agree with the broad idea of providing low-cost pragmatic approaches, I think it is important to consider the evidence base when making these recommendations. For example, different approaches are likely to be warranted for individuals

experiencing transient distress (which is common following adversity) compared to those who are showing signs of more severe psychopathology that may warrant intensive intervention. Approaches like Critical Incident Stress Debriefing have been demonstrated to be ineffective (or at worst, harmful) when indiscriminately applied in the aftermath of a traumatic event - these could potentially be conceptualized as ad-hoc group therapy sessions. I think this section could use some more consideration in terms of the specific interventions being recommended, ensuring that is specified who they would be recommended for, and that they are supported by research evidence.

Thank you for recommending this relevant nuance. We agree that there is no one-size-fits-all solution to psychopathology, and that it is important to highlight this fact particularly for cases of severe psychopathology. Therefore, we have added a section to the respective paragraph to clarify this point:

“However, we must note that we have not tested these direct implications for policy or clinical practice here, and the interventions mentioned are raised only as potential future implications. In addition, when administering or promoting such psychological interventions, it will always be important to consider the specific contexts and characteristics of the individuals, as well as the severity of their psychopathological symptoms. In particular, individuals who suffer from severe psychopathology following a major societal crisis (such as post-traumatic stress disorder) may require more intensive interventions such as individual psychotherapy. Indiscriminately applying solutions developed for more transient stress symptoms in these more severe cases might turn out to be ineffective or even detrimental to fostering psychological health³⁰.”

This is the study referred to with reference 30 above:

Litz, B. T. Early intervention for trauma: Where are we and where do we need to go? A commentary. *J Trauma Stress* **21**, 503–506; 10.1002/jts.20373 (2008).